# Influence of Perturbation's Type and Location on Treadmill Gait Regularity

Michalina Błażkiewicz [1],* and Anna Hadamus [2]

1  Faculty of Rehabilitation, The Józef Piłsudski University of Physical Education in Warsaw, 00-968 Warsaw, Poland
2  Department of Rehabilitation, Faculty of Dental Medicine, Medical University of Warsaw, 02-091 Warsaw, Poland; anna.hadamus@wum.edu.pl
*  Correspondence: michalina.blazkiewicz@awf.edu.pl

**Abstract: Background:** This study aimed to investigate how —external perturbations caused by a treadmill belt's acceleration (Acc) and deceleration (Dec) during the Initial-Contact (Initial), Mid-Stance (Mid), and Pre-Swing (ToeOff) phases affect gait regularity in young adults. **Methods:** Twenty-one healthy young females walked on a treadmill in a virtual environment (Motek GRAIL), in which four unexpected perturbations were applied to the left belt at the Initial, Mid, and ToeOff stages. Sample entropy (SampEn) was calculated for the center of mass (CoM) displacements for six perturbation scenarios in three directions—anterior–posterior (AP), medial–lateral (ML), and vertical (vert)—with SampEn vector lengths ($m$) ranging from 2 to 10. **Results:** The CoM displacement exhibited its highest regularity (low SampEn values) in the AP and vert directions during Dec–ToeOff, across all $m$ values. Similarly, this pattern was observed in the ML direction, but exclusively for $m = 2$ and 4. The least-regular CoM trajectories (high SampEn values) were for Dec–Mid in the AP direction, across all $m$ values. This trend persisted in the ML direction only for $m = 2$ and 4. However, the most irregular CoM displacements in the ML direction occurred during Dec–ToeOff for the remaining $m$ values. Vertical CoM displacements exhibited the highest irregularities during Dec–Initial for $m \geq 4$. **Conclusions:** Evaluating the regularity of CoM displacements using SampEn can be a useful tool for assessing how gait perturbations are handled.

**Keywords:** gait; perturbations; treadmill; provoked trips; provoked slips; entropy

## 1. Introduction

Gait perturbations refer to disruptions or irregularities in an individual's walking pattern or gait. These perturbations can occur due to various factors, including environmental obstacles [1], musculoskeletal conditions [2,3], neurological disorders [4], or external influences [5]. Environmental barriers include uneven terrain, path obstacles, and slippery surfaces. Musculoskeletal conditions involve muscle weakness, joint pain, or injuries, while neurological disorders (e.g., Parkinson's disease, multiple sclerosis, and stroke) can cause changes in posture and balance and, thus, the walking pattern. External influences like carrying heavy objects, wearing different footwear, or experiencing fatigue can also lead to gait perturbations. All of the above-mentioned factors can alter the way of walking and, most of all, the ability to maintain stability. According to Bruijn et al. [6], stable gait can be defined as a gait pattern that does not lead to falls, despite perturbations.

The literature provides several approaches for gait-stability assessment, generally divided into clinical and quantitative methods [6–8]. Clinical tests that assess postural and gait stability are the initial sources of knowledge about a patient's condition and can guide further diagnostic procedures. These tests evaluate functional characteristics, such as the time it takes to walk a distance safely and the ability to walk unassisted. These tests may include the timed 10 m walk test, the heel-to-toe test, the timed up and go (TUG) test, and the Babinski–Weil test [9]. Quantitative methods refer mainly to instrumented gait analysis,

which involves using specialized equipment such as force plates, motion capture systems, or wearable sensors. These tools provide detailed information on gait parameters, which are then analyzed to assess gait stability.

Clinical and quantitative gait-analysis methods offer distinct yet complementary perspectives when assessing gait stability. Clinical methods provide valuable insights into overall performance and are easily applicable in clinical settings. Quantitative methods (instrumented gait analysis) provide detailed data on joint angles, forces exerted, timing, and spatial parameters during walking. This quantitative approach offers a more comprehensive and in-depth understanding of gait mechanics and can detect subtle abnormalities that might not be evident through clinical observation alone. Integrating findings from both clinical and quantitative analyses allows for a more holistic evaluation and understanding of an individual's gait stability. The synergy between clinical and quantitative gait analysis methods enriches the assessment of gait stability by providing a multi-dimensional perspective that combines functional observations with detailed biomechanical insights. The above-mentioned methods are commonly employed to evaluate gait stability, providing fundamental data regarding an individual's behavior. However, it is essential to note that their analysis extends beyond basic measurements. Specifically, the manner in which these data are studied opens up an entirely different perspective, allowing for a comprehensive examination of how individuals respond to perturbations and maintain stability.

According to Hamacher et al. [10], both linear and nonlinear measures can be used to assess gait stability. Among the linear methods used to assess gait stability, the authors listed the following: step-and-stride length variability, step-and-stride time variability, double support time, gait speed, cadence, and the standard deviation of the analyzed time series. However, in recent years, a method associated with the margin-of-stability (MoS) determination has gained popularity [11]. Among the nonlinear measures, Hamacher et al. [10] listed wavelets and detrended fluctuation analysis, the fractal scaling index, Lyapunov exponents, and Floquet multipliers. Similar methods were mentioned in the review by Bruijn et al. [6]. It is worth noting that these two reviews [6,10] discussing the assessment of gait stability did not mention entropy, despite its recent popularity among nonlinear measures [12]. However, the papers mentioned focused on the possibility of predicting the probability of falling, and entropy does not have such predictive capabilities. Despite this, the use of entropy analysis in gait research has increased significantly over the past two decades [13].

Entropy analysis in gait research involves the application of principles from information theory to quantify the complexity, variability, and predictability within human gait patterns [14,15]. The concept of entropy, derived from thermodynamics and information theory, measures the randomness or disorder in a system. In the context of gait analysis, entropy-based approaches aim to characterize the complexity and regularity of human movement [13,15]. There are various entropy metrics used in gait research: approximate entropy (ApEn) [16], Shannon entropy [17], multiscale entropy (MSE) [17], and sample entropy (SampEn) [14,18].

ApEn measures the regularity and complexity of a time-series signal. It quantifies the unpredictability of fluctuations in gait patterns. Lower ApEn values suggest greater regularity or predictability, whereas higher values indicate more irregularity. Shannon entropy measures the uncertainty of average information content within a system. In gait research, it can quantify the diversity or variability of movement patterns across different conditions or populations. MSE examines entropy across multiple temporal scales, capturing how complexity changes over different timescales within a gait signal. It provides insights into both short-term and long-term dynamics of gait patterns. Like ApEn, SampEn quantifies the complexity/regularity/predictability/probability of analyzed motion [12,13]. The SampEn $(m, r, N)$ of a dataset of length $N$ measures the temporal pattern within the signal, assessing the logarithmic probability that two similar sequences having an identical number of data points $(m)$ will continue to be similar when an additional data point is introduced $(m + 1)$ without allowing self-matches [19]. Thus, the SampEn algorithm

necessitates three essential parameters: tolerance window $r$ (used to determine whether patterns within the time series are similar or not), vector length $m$ (data length compared across the time series to determine conditional probabilities), and time series length $N$. For gait data analysis, $m = 2, 4, 6, 8$ or $10$ is most often used, while $r$ should be equal to $0.2 \times SD$, where SD is the standard deviation of the analyzed time series [15,18,20,21]. An important fact is that SampEn offers an advantage in its independence from data length. It considers self-matches (repeated patterns) in the signal and is less sensitive to variations in data length compared to ApEn [22]. However, Richman and Moorman [22] recommended caution with datasets smaller than 200 points. Naturally, such short time series are unlikely in gait analysis when dealing with multiple gait cycles. Lower values of SampEn indicate greater regularity or predictability, which is associated with less complexity of structure [23]. Higher values suggest more complexity or randomness in the data. Given that complexity plays a crucial role in adapting to an environment, reduced complexity in physical movement results in diminished flexibility and increased rigidity in postural control [24]. Conversely, higher SampEn values (suggesting increased complexity) are interpreted as increased self-organization and an effective strategy in retaining postural control [25].

Applications of entropy analysis in gait research include characterizing gait variability, assessing gait stability, distinguishing healthy and pathological gaits, and understanding gait adaptation. In a topic describing the characterization of gait variability, entropy metrics help quantify irregularity and variability in gait patterns due to aging, injury, or neurological conditions [16]. In assessing gait stability, entropy analysis provides insight into gait stability and adaptability under various conditions or perturbations. Differences in entropy measures between healthy individuals and those with gait disorders or pathological conditions can aid in diagnostic or prognostic evaluations [26]. Moreover, entropy-based analysis can elucidate how individuals adapt their gait patterns in response to environmental changes or interventions [14].

The complex interaction between temporal and spatial control of joint motion during walking supports setting up stable gait patterns [27]. However, these patterns depend on factors such as age, pathology, and the specific type of perturbation involved. Older adults and individuals with pathologies often favor reduced variability and increased stability, adopting a slower gait. In contrast, healthy young adults show more variability, but remain more stable at different speeds [28,29]. However, the study of dynamic stability in response to sudden changes in speed remains an area that is lacking in comprehensive research. Park's study [14] is the only paper addressing a similar issue, examining how sudden changes in gait speed affect gait dynamics. It is worth noting that that study did not treat these changes as perturbations; instead, the speed changes (increase) occurred gradually, allowing the subjects time to adjust. However, to date, no studies have been conducted that comprehensively assess the stability, regularity, and complexity of gait among various perturbations involving both location and type of speed, including acceleration and deceleration. To address this gap, the primary aim of this study was to investigate how different perturbation possibilities, including their timing (Initial Contact, Mid-Stance, and Pre-Swing) and direction (acceleration or deceleration of the one belt of the treadmill), impact the regularity of the center of the body mass (CoM) movement. Since the regularity of time series is described using sample entropy, an additional purpose of this paper was closely related to the behavior of this parameter. Accordingly, the next objective was to examine the values of SampEn in the anterior–posterior (AP), medial–lateral (ML), and vertical directions, depending on the length of the vector $m$ for each type of perturbation possibility.

## 2. Materials and Methods

### 2.1. Participants

Twenty-one young women (age: 21.38 ± 1.32 years old; body weight: 61.38 ± 6.48 kg; and body height: 165.9 ± 4.53 cm) participated in this study. The participants met the following inclusion criteria: no muscular or neural diseases, no lower-limb injuries within

the six months preceding testing, and at least two days of activity per week as part of a physical recreation routine. Exclusion criteria included poor physical condition (evaluated subjectively on the day of the study), lack of experience in treadmill walking, and the use of medications that could adversely affect the nervous system.

This research followed the ethical guidelines and principles of the Declaration of Helsinki and received approval from the institutional review board of the Józef Piłsudski University of Physical Education in Warsaw, Poland (no. SKE01-15/2023). Informed consent was obtained from all participants before the study.

*2.2. Measurement Protocol and Perturbation Characteristics*

The kinematics and kinetics parameters of the perturbed gait were measured in a Gait Real-time Analysis Interactive Lab (GRAIL, Motek Medical B.V., Amsterdam, The Netherlands). The GRAIL is equipped with a dual split-belt treadmill (1000 Hz), a motion capture system (Vicon Metrics Ltd., Oxford, UK) (100 Hz), three video cameras, and synchronized virtual-reality environments. Participants' movements were captured using Human Body Model 2 (HBM2) with 26 reflective markers. Perturbation triggering and data acquisition were controlled using D-Flow software (Motek Medical B.V., Amsterdam, The Netherlands) [30].

In this study, the participants walked on the treadmill wearing flat-soled sports shoes at a constant speed of 1.2 m/s. As a safety precaution, each participant wore a safety harness connected to the ceiling, even though the perturbations were not intended to cause falls. Two types of unexpected perturbations were applied to the left belt of the treadmill, involving acceleration and deceleration (Figure 1). Each perturbation occurred at three specific points during the support phase of the gait cycle: Initial Contact, Mid-Stance, and Pre-Swing (Table 1). The magnitude of the perturbation was set at 5 on a scale of 1–5, involving a shift in treadmill belt speed of 0.5 m per second [4].

**Table 1.** Description of applied perturbations (the first column specifies the type of treadmill movement; the second column denotes the phase of the gait cycle in which the perturbation was applied; and the third column provides the corresponding label).

| Perturbation Type | Phase of the Gait Cycle | Labeling/ Perturbation Possibilities |
|---|---|---|
| Acceleration | Initial Contact<br>Mid-Stance<br>Pre-Swing | Acc–Initial<br>Acc–Mid<br>Acc–ToeOff |
| Deceleration | Initial Contact<br>Mid-Stance<br>Pre-Swing | Dec–Initial<br>Dec–Mid<br>Dec–ToeOff |

Each participant performed one trial for all six perturbation possibilities (type x phases). In each attempt, perturbations occurred at 10 s intervals on the left treadmill belt. Consequently, these perturbations appeared at the 30th, 40th, 50th, and 60th seconds of treadmill walking. These perturbations were applied to the left lower limb, as it was the supporting (non-dominant) leg for all subjects [31]. Leg dominance was determined using the kicking test, where participants indicated their preferred lower limb for kicking a ball.

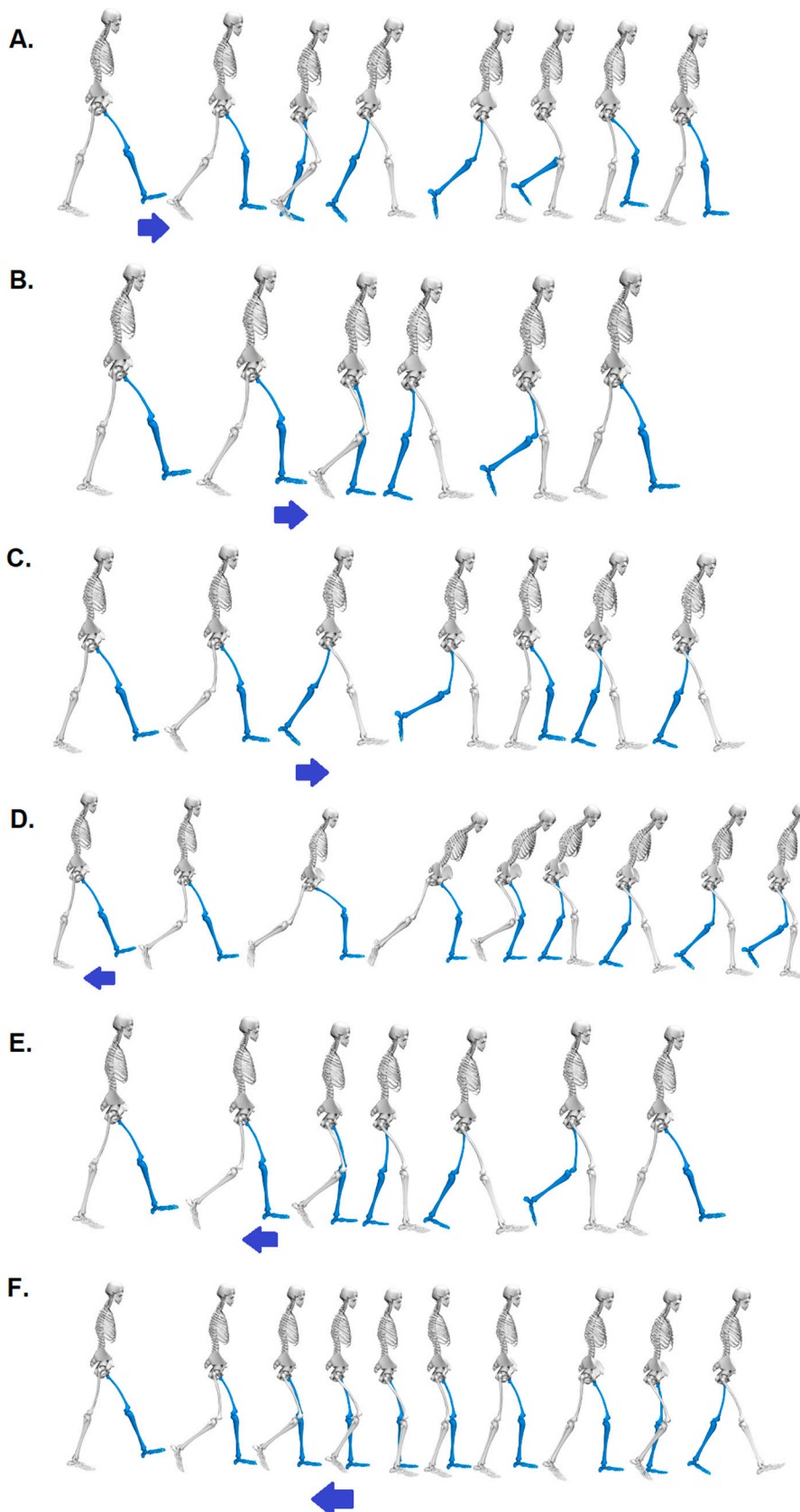

**Figure 1.** Visualization of the response to the six perturbation possibilities, including the key behavioral elements for treadmill acceleration (arrow pointing to the right) in phases—(**A**) Initial Contact, (**B**) Mid-Stance, (**C**) Pre-Swing—and for treadmill deceleration (arrow pointing to the left) in phases—(**D**) Initial Contact, (**E**) Mid-Stance, (**F**) Pre-Swing.

### 2.3. Time-Series Identification

For each of the six perturbation possibilities (Table 1), CoM displacements in three directions—anterior–posterior (AP), medial–lateral (ML) and vertical (vert), which included four perturbations—were considered for analysis. To ensure homogeneity of the data, the CoM time series were identified based on the vertical components of the ground reaction forces (GRF). The starting point of each time series was defined as the moment when the gait cycle involving the perturbation began (i.e., the contact of the heel with the ground of the left lower limb), and the endpoint was determined as the final gait cycle containing the response to the perturbation (indicated by the renewed contact of the left heel with the ground) (Figure 2). This treatment ensured that the raw data had the same length (3189 points).

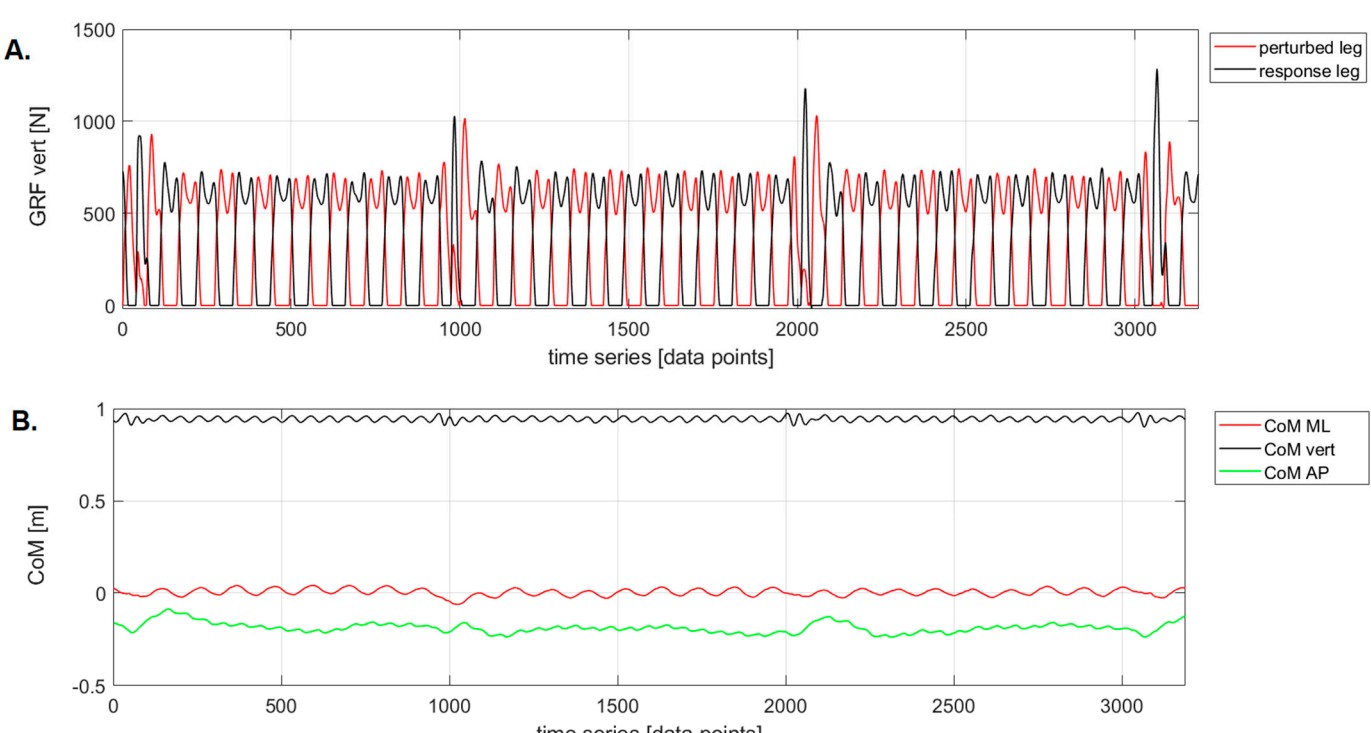

**Figure 2.** An example of raw data for treadmill acceleration during the Pre-Swing phase: (**A**) vertical ground reaction force trajectories (GRF–vert), (**B**) center of mass (CoM) trajectories for medial–lateral (ML), vertical (vert), and anterior–posterior (AP) displacements.

### 2.4. Sample Entropy Calculation

SampEn is defined as the negative natural logarithm of the conditional probability that a sequence of data points, each separated by a specified distance $m$, will repeat within a distance of $m + 1$, excluding self-matches:

$$\text{SampEn}(m, r, N) = -\ln\left(\frac{A^m(r)}{B^m(r)}\right)$$

where $B$ represents the total number of matches of length $m$. $A$ represents the subset of $B$ that also matches for $m + 1$. $N$ is the total number of data points in the time series, $m$ signifies the length of the vectors compared during data waveforms, and $r$ denotes the sensitivity threshold in which comparable vectors are considered similar.

Sample entropy was calculated for six perturbation possibilities (Table 1) on CoM–vert, CoM–AP, and CoM–ML signals using a code obtained from the Physionet tool [32]. The values chosen for $m$ were 2, 4, 6, 8, and 10, and $r$ was set to 0.2 times the mean standard

deviation of the time series ($r = 0.2 \times$ SD) [18,20,21]. The selection of $r = 0.2$ for the time series was based on the method proposed by Lake et al. [33].

In the case of human movement, a steady or periodic gait pattern has a low SampEn value, while a more complex gait pattern (a time series with large distances between data points) should have a higher SampEn value.

### 2.5. Statistical Analysis

The normality of the distribution of SampEn values, calculated for all $m$ values within six different perturbation possibilities, was assessed using the Shapiro–Wilk test. In most cases, the results indicated distributions that were different from normal. A Friedman ANOVA with Dunn–Bonferroni post hoc tests was used to assess the influence of $m$ levels on the SampEn, calculated for each of the six perturbation possibilities. Subsequently, a similar analysis was performed to identify which of the six perturbation possibilities and which direction exhibited the lowest gait regularity. These last two analyses were conducted independently for each $m$ level. Statistical analysis was performed using PQStat 2021 software version 1.8.2.238 (PQStat Software, Poznań, Poland). The level of significance was set at $p \leq 0.05$.

## 3. Results

The results are divided into three primary subsections. The first section investigated the impact of data length ($m = 2, 4, 6, 8, 10$) on SampEn values calculated for six types of perturbations (Table 1) by definition in the AP, ML, and vert directions. The values of the tolerance window $r$ were always equal to $0.2 \times$ SD, where SD was the standard deviation of the studied time series. This approach facilitated the identification of the specific $m$ value associated with the highest and lowest regularity across the six analyzed perturbation scenarios in each direction.

The subsequent section presents a comprehensive analysis, comparing regularities for time series calculated for six perturbations possibilities (Table 1)—separately, within each $m$ and each direction (AP, ML, vert).

The final chapter addresses an issue of which direction (AP, ML, or vert) exhibits the highest/lowest regularity within each of the six perturbations studied. As in earlier sections, the results are displayed for each $m$.

As demonstrated in the first subsection, the complexity of the analysis increased due to the impossibility of selecting a single optimal $m$. Nonetheless, this approach ensured a comprehensive examination of the data.

### 3.1. The Impact of Vector Length (m) on Sample Entropy Values across Six Perturbation Possibilities

After conducting Friedman's ANOVA with Dunn–Bonferroni post hoc tests, it was shown that for the ML direction (Figure 3A), for each of the six perturbation possibilities, significantly higher ($p < 0.0001$) SampEn values were noted for $m = 2$ than for those recorded for $m = 8, 10$. Additionally, significantly higher ($p < 0.0001$) SampEn values were found for $m = 4$ than for those calculated for $m = 6, 8$, and $10$, as well as for $m = 6$ in comparison to those noted for $m = 10$.

In the vertical direction (Figure 3B), the trend of stacking SampEn values was consistent with that described for the ML direction, with two exceptions. For $m = 2$, SampEn values were significantly higher than those calculated for $m = 6$, a difference not observed in the ML direction. For $m = 4$, SampEn values were not significantly higher than those reported for $m = 6$, which was noted in the ML direction.

In the AP direction, the trend of SampEn values did not follow a pattern from the previous directions (Figure 3C), illustrating remarkable diversity. For $m = 2$, SampEn values were significantly lower than those recorded for $m = 4, 6$, and $8$. Moreover, for $m = 6$, SampEn values were significantly higher than those associated with $m = 10$. These consistent trends extended across three perturbation possibilities (Acc–ToeOff, Acc–Mid,

and Dec–Mid). In the case of Acc–Initial, SampEn values at $m = 2$ were notably lower than those derived for $m = 6, 8,$ and 10. The identical relationship existed for Dec–ToeOff, except for relations involving $m = 10$. In contrast, for Dec–Initial, SampEn values at $m = 2$ were significantly lower when compared to all other SampEn counting possibilities.

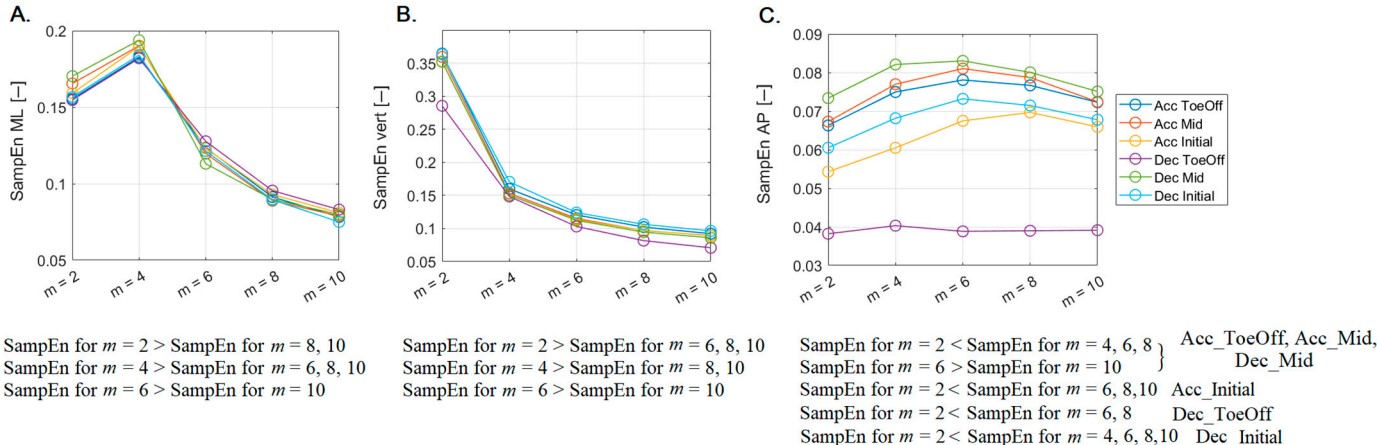

**Figure 3.** The impact of varying vector length ($m$) on median sample entropy (SampEn) values calculated for the center of mass (CoM) across different perturbations possibilities in (**A**) medial-lateral (ML), (**B**) vertical (vert), and (**C**) anterior–posterior (AP) directions. Statistical significance at the $p < 0.001$ level is shown at the bottom.

### 3.2. Impact of Perturbation Possibilities and m—Values on Gait Regularity within Directions

Friedman's ANOVA was performed within each combination of $m$ values ($m = 2, 4, 6, 8,$ and 10) and movement directions (AP, ML, and vert). The results of the Dunn–Bonferroni post hoc test for each direction are described below.

### 3.2.1. Medial–Lateral Direction (ML)

For $m = 2$ and 4, the SampEn values for Dec–Mid were the highest and significantly higher than those recorded for the Acc–ToeOff, Dec–ToeOff, and Dec–Initial (Figure 4).

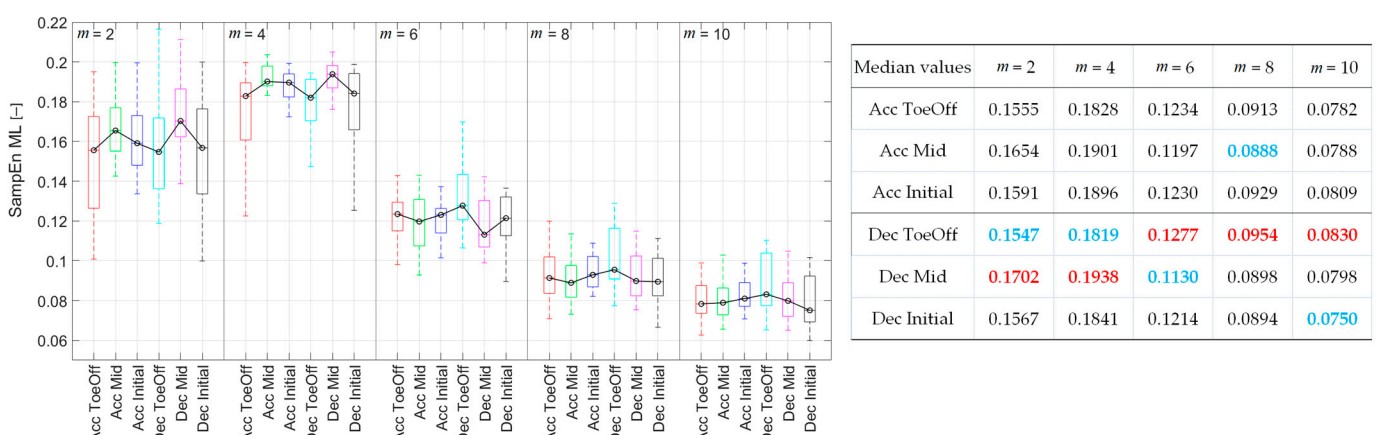

| Median values | $m = 2$ | $m = 4$ | $m = 6$ | $m = 8$ | $m = 10$ |
|---|---|---|---|---|---|
| Acc ToeOff | 0.1555 | 0.1828 | 0.1234 | 0.0913 | 0.0782 |
| Acc Mid | 0.1654 | 0.1901 | 0.1197 | 0.0888 | 0.0788 |
| Acc Initial | 0.1591 | 0.1896 | 0.1230 | 0.0929 | 0.0809 |
| Dec ToeOff | 0.1547 | 0.1819 | 0.1277 | 0.0954 | 0.0830 |
| Dec Mid | 0.1702 | 0.1938 | 0.1130 | 0.0898 | 0.0798 |
| Dec Initial | 0.1567 | 0.1841 | 0.1214 | 0.0894 | 0.0750 |

**Figure 4.** Sample entropy values for each perturbation possibility in the medial–lateral direction as a function of the value of $m$. On each box, the central mark indicates the median and the bottom and top edges of the box indicate the 25th and 75th percentiles, respectively. The whiskers extend to the most extreme data points. In the table, blue values indicate the minimal sample entropy values, and red values indicate the maximal ones.

For $m = 6$, the effect of treadmill deceleration in the Pre-Swing phase (Dec–ToeOff) caused the highest CoM irregularity. The SampEn value for this perturbation possibility

was significantly higher than those values recorded for the Acc–Mid and Dec–Mid. No significant differences in SampEn values were observed for $m = 8$ and 10.

### 3.2.2. Vertical Direction (Vert)

For $m = 2$, the CoM signal exhibited the highest regularity during Dec–ToeOff. The SampEn value in this phase was significantly lower than those for the other five perturbations (Figure 5). This pattern persisted for $m = 6$, 8, and 10, but only for two perturbations. The CoM irregularity was significantly higher for Acc–ToeOff and Dec–Initial than those observed for Dec–ToeOff. Additionally, for $m = 10$, significantly higher SampEn values were recorded for Acc–Initial and Dec–Mid than for Dec–ToeOff. In the case of $m = 2$, there were significantly higher SampEn values for Acc_–Mid than for Dec–ToeOff. For $m = 4$, SampEn values behaved differently, indicating the highest values for Dec–Initial. The values for this perturbation were significantly higher than those recorded for Acc–Initial, Dec–ToeOff, and Dec–Mid. In summary, the CoM displacement achieved its highest regularity, indicated by low SampEn values during Dec–ToeOff, across all $m$ values. The lowest regularity was noted during Dec–Initial for $m = 4$, 6, 8, and 10.

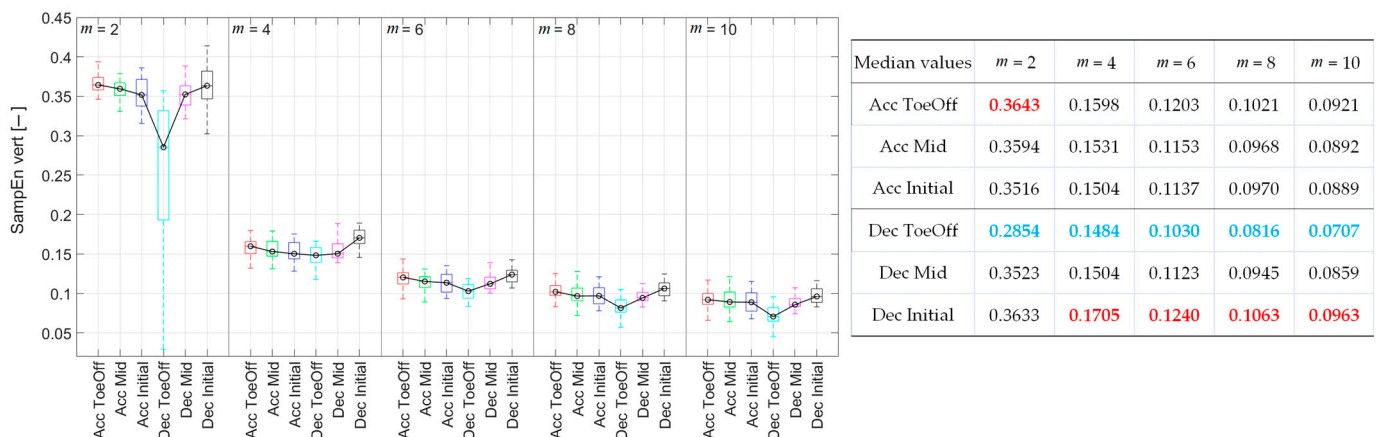

| Median values | $m = 2$ | $m = 4$ | $m = 6$ | $m = 8$ | $m = 10$ |
|---|---|---|---|---|---|
| Acc ToeOff | 0.3643 | 0.1598 | 0.1203 | 0.1021 | 0.0921 |
| Acc Mid | 0.3594 | 0.1531 | 0.1153 | 0.0968 | 0.0892 |
| Acc Initial | 0.3516 | 0.1504 | 0.1137 | 0.0970 | 0.0889 |
| Dec ToeOff | 0.2854 | 0.1484 | 0.1030 | 0.0816 | 0.0707 |
| Dec Mid | 0.3523 | 0.1504 | 0.1123 | 0.0945 | 0.0859 |
| Dec Initial | 0.3633 | 0.1705 | 0.1240 | 0.1063 | 0.0963 |

**Figure 5.** Sample entropy values for each perturbation possibility in the vertical direction for various $m$. On each box, the central mark indicates the median and the bottom and top edges of the box indicate the 25th and 75th percentiles, respectively. The whiskers extend to the most extreme data points. In the table, blue values indicate the minimal sample entropy values, and red values indicate the maximal ones.

### 3.2.3. Anterior–Posterior Direction (AP)

Differently from the previous directions, for $m = 2$ and 4, it was shown that there were no statistically significant differences in the regularity of CoM displacement due to individual perturbations. For the remaining $m$ values, it was demonstrated that treadmill accelerations during the Initial Contact, Mid-Stance, and Pre-Swing phases led to a significant increase in the irregularity of CoM displacement compared to that observed for Dec–ToeOff. Additionally, for $m = 8$ and 10, a significant increase in SampEn value was observed for Dec–Mid perturbations compared to Dec–ToeOff perturbations (Figure 6). In summary, the CoM displacement exhibited its highest regularity, as indicated by low SampEn values during Dec–ToeOff, and the lowest regularity, as indicated by high SampEn values, during Dec–Mid, across all $m$ values.

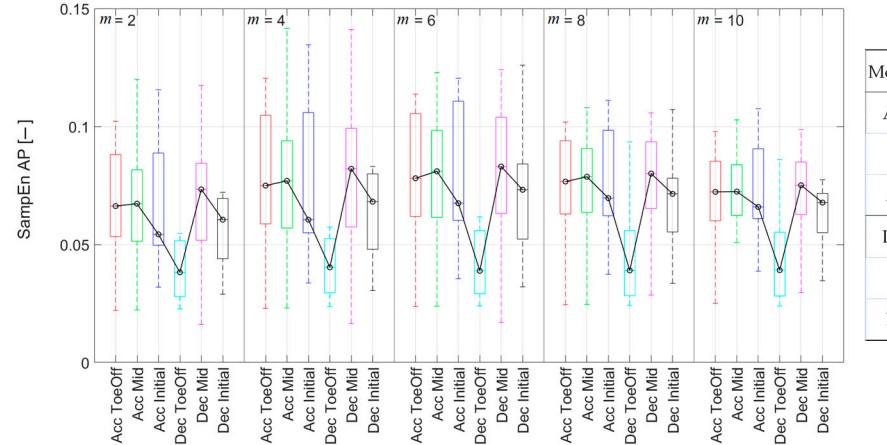

| Median values | $m = 2$ | $m = 4$ | $m = 6$ | $m = 8$ | $m = 10$ |
|---|---|---|---|---|---|
| Acc ToeOff | 0.0662 | 0.0749 | 0.0780 | 0.0766 | 0.0723 |
| Acc Mid | 0.0672 | 0.0769 | 0.0810 | 0.0787 | 0.0724 |
| Acc Initial | 0.0543 | 0.0605 | 0.0674 | 0.0696 | 0.0659 |
| Dec ToeOff | 0.0382 | 0.0403 | 0.0388 | 0.0389 | 0.0391 |
| Dec Mid | 0.0733 | 0.0820 | 0.0830 | 0.0800 | 0.0751 |
| Dec Initial | 0.0605 | 0.0681 | 0.0732 | 0.0714 | 0.0677 |

**Figure 6.** Sample entropy values for each perturbation possibility in the anterior–posterior direction as a function of the value of *m*. On each box, the central mark indicates the median, and the bottom and top edges of the box indicate the 25th and 75th percentiles, respectively. The whiskers extend to the most extreme data points. In the table, blue values indicate the minimal sample entropy values, and red values indicate the maximal ones.

### 3.3. Effect of Direction on Gait Regularity across Different m-Values

Examining the impact of direction (AP, ML, and vert) on the regularity of the CoM signal within different *m* values, it was shown that for the perturbations related to treadmill acceleration in the Pre-Swing phase (Figure 7A), there was significantly higher irregularity of the CoM signal in the ML and vert directions compared to that noted for the AP direction. This pattern held for *m* = 2, 4, and 6. Additionally, for *m* = 2, the SampEn value in the ML direction was significantly lower than that in the vertical direction. No statistically significant differences were found for the other *m* values.

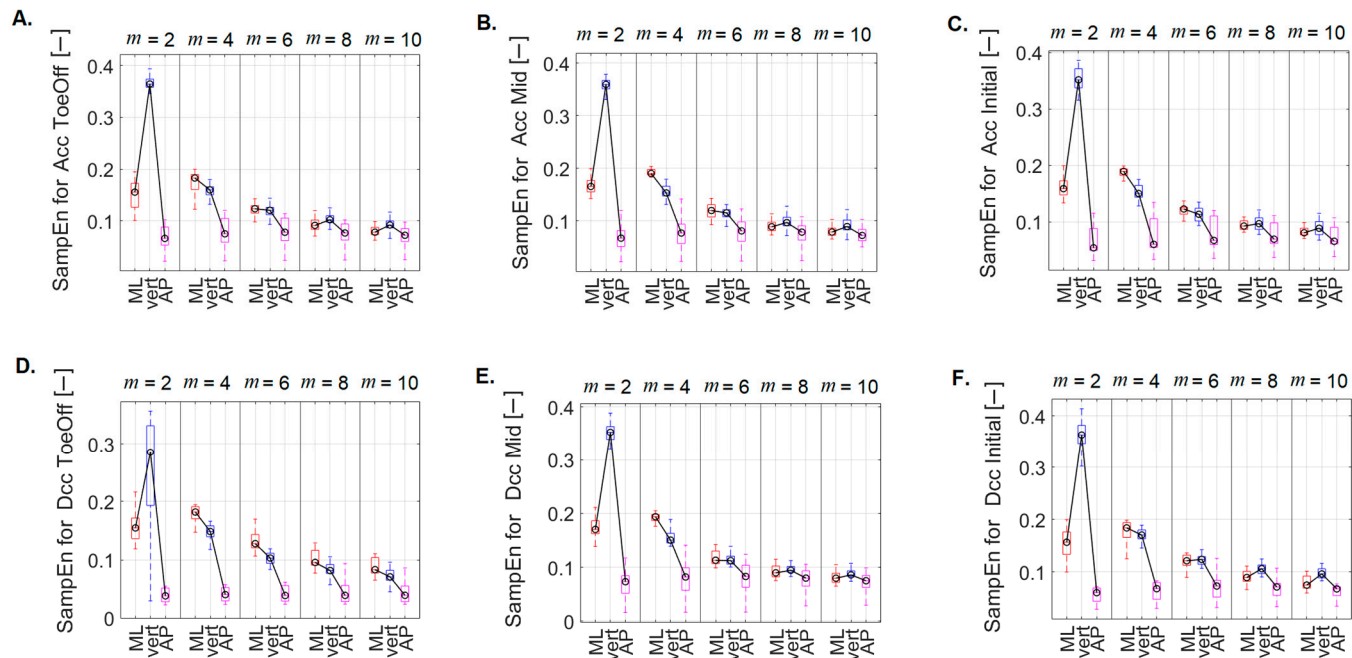

**Figure 7.** Differences between SampEn values calculated for *m* = 2, 4, 6, 8, and 10 for comparable directions (ML, vert, and AP) for each of the six perturbation possibilities: (**A**) Acc–ToeOff, (**B**) Acc–Mid, (**C**) Acc–Initial, (**D**) Dec–ToeOff, (**E**) Dec–Mid, (**F**) Dec–Initial. On each box, the central mark indicates the median and the bottom and top edges of the box indicate the 25th and 75th percentiles, respectively. The whiskers extend to the most extreme data points.

For the perturbations related to treadmill acceleration in the Mid-Stance phase (Figure 7B) and for $m$ = 2, 4, and 6, notably higher SampEn values were in the ML and vert directions than those recorded in the AP direction. Furthermore, for $m$ = 2, significantly higher SampEn values were recorded in the vert direction than those in the ML direction. For $m$ = 4, this relationship was reversed. Additionally, for $m$ = 10, significantly higher SampEn values were recorded in the vert direction than those in the AP direction.

For the perturbations related to treadmill acceleration in the Initial Contact phase (Figure 7C) and for $m$ = 2, 4, significantly higher SampEn values were in the ML and vert directions than those noted for the AP direction. Furthermore, for $m$ = 2, the SampEn value in the vert direction was significantly higher than that recorded in the ML direction. For $m$ = 4, SampEn–ML was greater than SampEn–vert. For $m$ = 6, SampEn–ML was greater than SampEn–AP, while for $m$ = 10, SampEn–vert exceeded SampEn–AP. As with the previous perturbations, no statistically significant differences were found for $m$ = 8.

For treadmill deceleration in the Pre-Swing phase (Figure 7D), with $m$ = 2, 4, 8, and 10, significantly higher SampEn values were in the ML and vert directions than those noted for the AP direction. For $m$ = 6, significantly higher SampEn values were in the ML direction versus those in the vert and AP directions.

For a perturbation containing treadmill deceleration in the Mid-Stance phase (Figure 7E), for $m$ = 2, 4, and 6, significantly higher SampEn values were in the ML and vert directions versus those recorded for the AP direction. Additionally, for $m$ = 2, SampEn–vert was greater than SampEn–ML. On the other hand, for $m$ = 4, this relationship was opposite. For $m$ = 8 and 10, SampEn–vert was greater than SampEn–AP.

For a perturbation containing treadmill lag in the initial contact phase (Figure 7F), for $m$ = 2, 4, 6, and 8, significantly higher SampEn values were in the ML and vert directions compared to those recorded for the AP direction. In addition, for $m$ = 2 and 8, SampEn–vert was greater than SampEn–ML. For $m$ = 10, SampEn–vert was greater than SampEn–ML, and SampEn–AP.

## 4. Discussion

The primary aim of this study was to investigate the impact of different perturbation possibilities, including their timing (Initial Contact, Mid-Stance, and Pre-Swing) and direction (acceleration or deceleration of the one treadmill belt), on gait regularity. Gait regularity was assessed based on the displacement of the center of mass (CoM) in the anterior–posterior (AP), medial–lateral (ML), and vertical (vert) directions. As the regularity of the CoM time series was quantified using sample entropy (SampEn), an additional aim of this study was closely associated with understanding the behavior of this parameter. Consequently, the subsequent aim was to analyze SampEn values in the above-mentioned directions, considering the comparison of the vector length ($m$).

This study implemented a perturbation protocol via the GRAIL system (Motek Medical BV, Amsterdam, The Netherlands), enabling precise timing and intensity control of perturbations. This accuracy facilitated the manipulation of the time-series duration for analysis. Although the SampEn value is independent of the time series length, Richman and Moorman [22] recommended that the analyzed data consist of at least 200 points. In this study, each dataset comprised 3189 points. However, the SampEn value depends on the choice of $m$ and $r$ parameters, and determining suitable parameters for gait analysis is not straightforward [34]. Studies focusing on postural control during quiet standing commonly use an $m$ value of 2, while $r$ equals 0.2 times the standard deviation of the data (SD) [12]. However, when considering gait analysis, the selection of parameters becomes less clear, particularly due to the authors' focus on various parameters, such as joint angle and torque waveforms, as well as stride characteristics [35,36]. According to Tochigi et al. [37], employing a longer $m$ template (higher $m$ value) theoretically augments specificity in identifying matched templates, but may lead to a potential decrease in discriminatory power. Many researchers [34,37] have opted for a minimum template length ($m$ = 2) for gait-related data to maximize discriminatory power. Additionally, previous authors [15,18,20] explored

higher *m* values (*m* = 4, 6, 8, 10), resulting in consistently lower SampEn values, a trend also observed in this study, mainly for SampEn calculated in ML and vert direction. For the AP direction, the trend differed—SampEn values were the lowest for *m* = 2 across all perturbations, except for treadmill deceleration during the Pre-Swing phase. In this case, as *m* values increased, SampEn values decreased.

To date, no studies have been found that describe the regularity of the human body movement or its segments in reaction to gait perturbations using nonlinear measures, particularly sample entropy. This paper presented results regarding the SampEn values computed for the CoM time series separately in the anterior–posterior (AP), medial–lateral (ML), and vertical directions. This approach enabled the identification of perturbations causing the most irregular gait. It is important to note that the analysis was conducted for *m* equal to 2, 4, 6, 8, and 10. The study revealed clear patterns in the center of mass (CoM) displacement, particularly in the vertical and anterior–posterior directions. The most coherent CoM displacement, indicated by low SampEn values indicating regularity, occurred during treadmill deceleration in the Pre-Swing phase. This consistency was reproducible across all *m* in both AP and vertical directions. This coherence suggests that the body anticipated and skillfully adapted to this perturbation, recovering quickly to the baseline displacement of CoM, resulting in a regular gait. However, higher SampEn values, reflecting increased irregularity, were recorded at other times, depending on the direction. In the AP direction, irregular CoM behavior was noticeable during treadmill deceleration in the Mid-Stance phase, indicated by the highest SampEn values across all *m*-values. Conversely, in the vertical direction, the delay of the treadmill during the Initial Contact phase exhibited the highest SampEn values (for *m* = 4, 6, 8, and 10), indicating irregularity. For *m* = 2, the highest irregularity occurred during treadmill acceleration in the Pre-Swing phase.

Regarding the medial–lateral direction, the highest irregularity in CoM displacement was observed due to treadmill belt deceleration during the Mid-Stance phase, specifically for *m* = 2 and 4. However, other *m* values increased CoM irregularity during treadmill deceleration in the Pre-Swing phase. It is important to note that when examining extreme values, the highest irregularity (maximum value of SampEn) was observed for the CoM time series involving acceleration-type perturbations during the Pre-Swing phase, calculated for *m* = 2 in the vertical direction. Conversely, the highest order (minimum value of SampEn) was for the CoM signal for the anterior–posterior direction during the deceleration perturbation in the Pre-Swing phase (also calculated for *m* = 2). According to a study by Sloot et al. [4], the above results are adequately confirmed. Sloot et al. [4] were among the few researchers who conducted experiments utilizing identical equipment and settings, operating under conditions similar to those presented in this paper (perturbation intensity and treadmill-based accelerations and decelerations). The authors demonstrated that the applied perturbations appeared to have a limited effect on the gait pattern. Even at the highest intensities, the observed impact resulted in less than a 4° knee flexion or extension. Furthermore, there was a decrease of less than 4% in stride length and less than a 3.5% change in both stride time and stance phase, some of which were inherent to the perturbation itself. Notably, during deceleration, there was a 6.1% increase in step width. This adjustment might indicate an attempt to improve the base of support in the medial–lateral direction, compensating for the decrease in the anterior–posterior direction resulting from the deceleration [38]. In contrast, during acceleration, step width remained unchanged, suggesting that the subjects were not affected by the perturbations.

This study had several limitations, which also contribute value by showing the way for future research directions to complement the current approach. The first limitation refers to the way perturbations were triggered. They were applied solely on the left belt of the treadmill and occurred at intervals of every 10 s. The effect on gait regularity resulting from perturbations applied to the right treadmill lane or at different frequencies remains unclear. Consequently, in the case of young subjects, this setup allowed them to return to a typical gait pattern. Another limitation concerned the methodology used to calculate

sample entropy. While different $m$ values were considered, tolerance window $r$ was not addressed. Ahmadi et al. [18] proposed a range of 0.1 to 0.3 times the standard deviation as potential $r$ values. Unfortunately, this approach could significantly complicate the overall analysis. Additionally, a limitation exists in the exclusive analysis of the regularity of center of mass (CoM) displacements, while exploring trunk behavior might offer a more accurate determination. Readers may discover further limitations beyond those mentioned here; however, these will continually serve as new directions and sources of inspiration to enhance and complete this study and its approach.

## 5. Conclusions

Based on the observed patterns in the regularity of CoM displacements across gait perturbations, several conclusions can be drawn.

First, during the treadmill deceleration in the Pre-Swing phase, the CoM displacement exhibited its most consistent pattern (indicated by low sample entropy values) in the AP and vertical directions across all $m$ values. Conversely, the least regular CoM trajectories (with high SampEn values) were evident during treadmill deceleration in Mid-Stance in the AP direction, consistent across all $m$ values, and were also observed in the ML direction, particularly for $m = 2$ and 4.

These findings highlight the significance of assessing the regularity of CoM movements using SampEn as a valuable tool in comprehending the management and response to gait perturbations. They shed light on how different perturbations influence the regularity of CoM movements in various directions, offering insight into the adaptability and stability of the human gait under perturbed conditions.

**Author Contributions:** Conceptualization, M.B.; methodology, M.B.; software, M.B.; validation, M.B. and A.H.; formal analysis, M.B.; investigation, M.B.; resources, M.B.; data curation, M.B.; writing—original draft preparation, M.B.; writing—review and editing, M.B. and A.H.; visualization, M.B. and A.H.; supervision, M.B.; project administration, M.B.; funding acquisition, M.B. and A.H. All authors have read and agreed to the published version of the manuscript.

**Funding:** This research was funded by the Józef Piłsudski University of Physical Education in Warsaw, grant number UPB no. 2 (114/12/PRO/2023). Gait Real-time Analysis Interactive Lab (GRAIL, Motek Medical B.V., Amsterdam, The Netherlands) was purchased under the project "Adaptation and equipment of innovative laboratories for diagnostic and therapeutic tests of the musculoskeletal system" co-financed by the European Union under the Operational Program Development of Eastern Poland 2007–2013. Apparatus was maintained under a subjective subsidy from the Ministry of Education and Science under decision 41/529869/SPUB/SP/2022.

**Institutional Review Board Statement:** The study was conducted in accordance with the Declaration of Helsinki and approved by the Institutional Review Board of Józef Piłsudski University of Physical Education in Warsaw, Poland (protocol code SKE01-15/2023; date of approval 24 March 2023).

**Informed Consent Statement:** Informed consent was obtained from all subjects involved in the study. Written informed consent was obtained from the patients to publish this paper.

**Data Availability Statement:** The measurement data used to support the findings of this study are available from the corresponding author upon request. The data are not publicly available due to privacy concerns and the fact that it is part of an ongoing study.

**Acknowledgments:** We would like to thank Karol Kowieski, Katarzyna Kaczmarczyk, Martyna Jarocka, and Agnieszka Zdrodowska for their help in data collection.

**Conflicts of Interest:** The authors declare no conflict of interest.

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
