# Peer review of "Influence of Perturbation’s Type and Location on Treadmill Gait Regularity"

_applsci, doi:10.3390/app14020493_

Round 1

Reviewer 1 Report

Comments and Suggestions for Authors

Dear authors,

The article provides a comprehensive overview of gait perturbations, outlining various factors that can contribute to disruptions in an individual's walking pattern. There are points to improve in your manuscript:

1. The article provides a comprehensive overview of gait perturbations, outlining various factors that can contribute to disruptions in an individual's walking pattern. The article effectively introduces the concept of gait stability and highlights the distinction between clinical and quantitative assessment methods. The mention of clinical tests such as the Timed 10-meter walk test and quantitative methods involving instrumented gait analysis contributes to the overall clarity of the introduction. However, it would be beneficial to briefly explain how these methods complement each other in assessing gait stability, offering a smoother transition between the two approaches.

2.      The introduction effectively establishes the context and importance of studying gait perturbations, provides a solid background on assessment methods, and outlines the study's objectives. However, addressing the gap in the literature regarding the entropy in previous reviews and further elaborating on the integration of clinical and quantitative methods could enhance the overall coherence of the introduction.

3.      In the results section, a notable challenge arises in deciphering the findings due to a lack of clarity in the presentation. The absence of detailed explanations, statistical indicators, and contextualization of results within the broader research landscape hinders the reader's comprehension.

4.      In the discussion section, the limitations are inadequately addressed, the absence of a discussion on future directions, and potential biases inherent in the research design and execution.

Overall, while the introduction successfully establishes the importance of the research topic, the article could benefit from a clearly defined study design. Additionally, a deeper integration of relevant literature, a clearer presentation of results, and a more comprehensive discussion of limitations and future directions would undoubtedly elevate the merit of the article.

Author Response

Reviewer 1:

Dear authors,

The article provides a comprehensive overview of gait perturbations, outlining various factors that can contribute to disruptions in an individual's walking pattern. There are points to improve in your manuscript:

  1. The article provides a comprehensive overview of gait perturbations, outlining various factors that can contribute to disruptions in an individual's walking pattern. The article effectively introduces the concept of gait stability and highlights the distinction between clinical and quantitative assessment methods. The mention of clinical tests such as the Timed 10-meter walk test and quantitative methods involving instrumented gait analysis contributes to the overall clarity of the introduction. However, it would be beneficial to briefly explain how these methods complement each other in assessing gait stability, offering a smoother transition between the two approaches.

Thank you for this suggestion. The paper has been updated accordingly.

  1. The introduction effectively establishes the context and importance of studying gait perturbations, provides a solid background on assessment methods, and outlines the study's objectives. However, addressing the gap in the literature regarding the entropy in previous reviews and further elaborating on the integration of clinical and quantitative methods could enhance the overall coherence of the introduction.

We appreciate your suggestion, and we've made an effort to improve this section of the introduction.

  1. In the results section, a notable challenge arises in deciphering the findings due to a lack of clarity in the presentation. The absence of detailed explanations, statistical indicators, and contextualization of results within the broader research landscape hinders the reader's comprehension.

Thank you. We hope that clarification provided at the start of the results section will enhance comprehension. The complexity of the entire analysis arises from the challenge of selecting a single 'm' value for Sample Entropy calculation, resulting in fluctuating entropy values, as demonstrated. Regrettably, numerous authors have a tendency to ignore this aspect and solely compute for m = 2, which is suitable for evaluating postural control but only in the context of standing (also not always). Consequently, this approach shortens the analysis. 

  1. In the discussion section, the limitations are inadequately addressed, the absence of a discussion on future directions, and potential biases inherent in the research design and execution.

Thank you for your comment. We have included the necessary paragraph.

Overall, while the introduction successfully establishes the importance of the research topic, the article could benefit from a clearly defined study design. Additionally, a deeper integration of relevant literature, a clearer presentation of results, and a more comprehensive discussion of limitations and future directions would undoubtedly elevate the merit of the article.

Thank you for your insightful feedback. We acknowledge the strengths highlighted in the introduction regarding the research topic's importance. We take into account the suggestions provided to enhance the article further, focusing on refining the study design for clearer comprehension. Additionally, we improved the presentation of results for greater clarity, and provided a more comprehensive discussion of limitations and future directions to enhance the overall merit and value of the article.

Reviewer 2 Report

Comments and Suggestions for Authors

In the intro, the second paragraph described by the author seems to have no theme. I suggest that the author divides this paragraph into two parts and confirms the narrative theme of each paragraph.

The description of SampEn needs to be more detailed.

The research hypothesis needs to be added in the last paragraph.

There are three main issues in the methods section.

1.Why are perturbations only set on the left blet? This study does not only focus on gait changes(length, speed et al.), but more on the center of mass. So why design the experiment this way?

2. The experimental flow chart cannot restore the experimental design at all, and this part needs to be redone.

3. During the experiment, was the subject’s trunk state consistent every time he responded to perturbations? According to common sense, it should be impossible. So how does the author deal with this problem?

Author Response

Reviewer 2

In the intro, the second paragraph described by the author seems to have no theme. I suggest that the author divides this paragraph into two parts and confirms the narrative theme of each paragraph.

Thank you for the suggestion. We have made significant improvements of this section. Now, it is divided into three sections:

  1. Definition of gait perturbation and factors influencing it.
  2. Approaches for gait stability assessment categorized into clinical and quantitative methods with detailed descriptions.
  3. Gait stability assessment methods covering both linear and nonlinear approaches.

This version provides a clear overview of the revised structure of the introduction, highlighting the sections and their respective content.

 The description of SampEn needs to be more detailed.

Thank you, this part has been corrected.

The research hypothesis needs to be added in the last paragraph.

Thank you sincerely for your suggestion. However, creating a hypothesis becomes challenging when the study and its analytical method deviate significantly from existing studies. Under these circumstances, generating hypotheses would involve contrived assumptions that lack authenticity. Hypotheses typically stem from insights or suspicions grounded in existing literature. In this instance, due to the absence of comparable papers, we didn't possess any preconceived notions or intuitions to formulate hypotheses.

 There are three main issues in the methods section.

  1. Why are perturbations only set on the left belt? This study does not only focus on gait changes (length, speed et al.), but more on the center of mass. So why design the experiment this way?

This experiment is a segment of a larger project involving two distinct age groups: children aged 7-10 and the elderly over 65. As the left limb serves as the supporting limb in all participants, we opted for perturbations on the left treadmill belt for safety reasons. Additionally, this article addresses only one aspect of our inquiries. Other questions include how the distribution of muscle forces changes and whether all individuals respond uniformly to the applied perturbations, among many others.

  1. The experimental flow chart cannot restore the experimental design at all, and this part needs to be redone.

Thank you for this remark. We lack precise insight into the specific aspects that are difficult to understand, making it challenging to improve this section. Furthermore, when delineating the data collection process using the Grail system, our approach was guided by cited papers known for their clear and comprehensive presentation of this procedure.

Sloot, L.H.; van den Noort, J.C.; van der Krogt, M.M.; Bruijn, S.M.; Harlaar, J. Can Treadmill Perturbations Evoke Stretch Reflexes in the Calf Muscles? PLOS ONE 2015, 10, e0144815, doi:10.1371/journal.pone.0144815

Schrijvers, J.C.; van den Noort, J.C.; van der Esch, M.; Harlaar, J. Responses in knee joint muscle activation patterns to different perturbations during gait in healthy subjects. Journal of Electromyography and Kinesiology 2021, 60, 102572, doi:https://doi.org/10.1016/j.jelekin.2021.102572.

  1. During the experiment, was the subject’s trunk state consistent every time he responded to perturbations? According to common sense, it should be impossible. So how does the author deal with this problem?

During the experiment, the subject's trunk state did not remain consistent every time they responded to perturbations. Common sense dictates this inconsistency to be expected. However, it's important to note that the objective of the study wasn't solely to analyze trunk displacement but rather focused on the body's center of gravity, typically situated around the 2nd sacral vertebra in a healthy individual. The function of the trunk itself is a separate aspect that wasn't the primary focus. Thank you for your comment; we appreciate it and plan to address this aspect in a forthcoming article.

Round 2

Reviewer 2 Report

Comments and Suggestions for Authors

none